# The structure of the Ctf19c/CCAN from budding yeast

Stephen M Hinshaw*, Stephen C Harrison*

Department of Biological Chemistry and Molecular Pharmacology, Harvard Medical School, Howard Hughes Medical Institute, Boston, United States

**Abstract** Eukaryotic kinetochores connect spindlemicrotubules to chromosomal centromeres. A group of proteins called the Ctf19 complex (Ctf19c) in yeast and the constitutive centromere associated network (CCAN) in other organisms creates the foundation of a kinetochore. The Ctf19c/CCAN influences the timing of kinetochore assembly, sets its location by associating with a specialized nucleosome containing the histone H3 variant Cse4/CENP-A, and determines the organization of the microtubule attachment apparatus. We present here the structure of a reconstituted 13-subunit Ctf19c determined by cryo-electron microscopy at ~4 Å resolution. The structure accounts for known and inferred contacts with the Cse4 nucleosome and for an observed assembly hierarchy. We describe its implications for establishment of kinetochores and for their regulation by kinases throughout the cell cycle.

## Introduction

During cell division, depolymerizing microtubules pull copies of the genome into developing daughter cells. The pulling force is transmitted to the centromere through the kinetochore, the apparatus that determines the position of the microtubule contact point along the chromosome, couples chromosome movements to microtubule dynamics, organizes and responds to the activities of kinases, and establishes a chromosomal domain that promotes proper microtubule attachment (*Biggins, 2013*; *Hinshaw and Harrison, 2018*; *McKinley and Cheeseman, 2016*). The kinetochore is an assembly of biochemically distinct subcomplexes (*Cheeseman et al., 2002*; *De Wulf et al., 2003*), and coordinated appearance and disappearance of these subcomplexes over evolutionary timescales suggests a functional modularity (*van Hooff et al., 2017*). One such assembly is the Ctf19c/CCAN, which anchors the kinetochore on chromosomal DNA.

Kinetochore proteins assemble on specialized nucleosomes in which Cse4/CENP-A replaces histone H3. In animals, at least two conserved kinetochore proteins, CENP-C and CENP-N, confer specificity for CENP-A (*Carroll et al., 2010*; *Carroll et al., 2009*). The vertebrate CCAN also stabilizes CENP-A nucleosomes so that they are maintained throughout the cell cycle and during extended periods of cellular quiescence (*Cao et al., 2018*; *Guo et al., 2017*; *Smoak et al., 2016*). The protein domains of CENP-C and CENP-N that confer CENP-A selectivity are present in yeast Mif2 and Chl4, respectively, indicating that the mechanism of CENP-A/Cse4 nucleosome recognition is probably conserved. In addition to recognizing Cse4 and supporting outer kinetochore assembly, Ctf19c proteins couple cohesin recruitment with DNA replication initiation and direct successful meiotic chromosome segregation (*Hinshaw et al., 2017*; *Marston et al., 2004*; *Vincenten et al., 2015*).

The first Ctf19c factors were identified due to overlapping functions in chromosome transmission fidelity (reviewed in *Hinshaw and Harrison, 2018*). The yeast proteins, like their human orthologs, co-purify from cells and depend on each other for kinetochore localization (*Cheeseman et al., 2002*; *De Wulf et al., 2003*; *Foltz et al., 2006*; *Klare et al., 2015*; *Okada et al., 2006*; *Pekgöz Altunkaya et al., 2016*; *Weir et al., 2016*). There are 13 Ctf19c subunits (*Figure 1A*, *Table 1*, *Pekgöz Altunkaya et al., 2016*), and these associate with a second complex containing Mif2/

*For correspondence:
hinshaw@crystal.harvard.edu (SMH);
harrison@crystal.harvard.edu (SCH)

Competing interests: The authors declare that no competing interests exist.

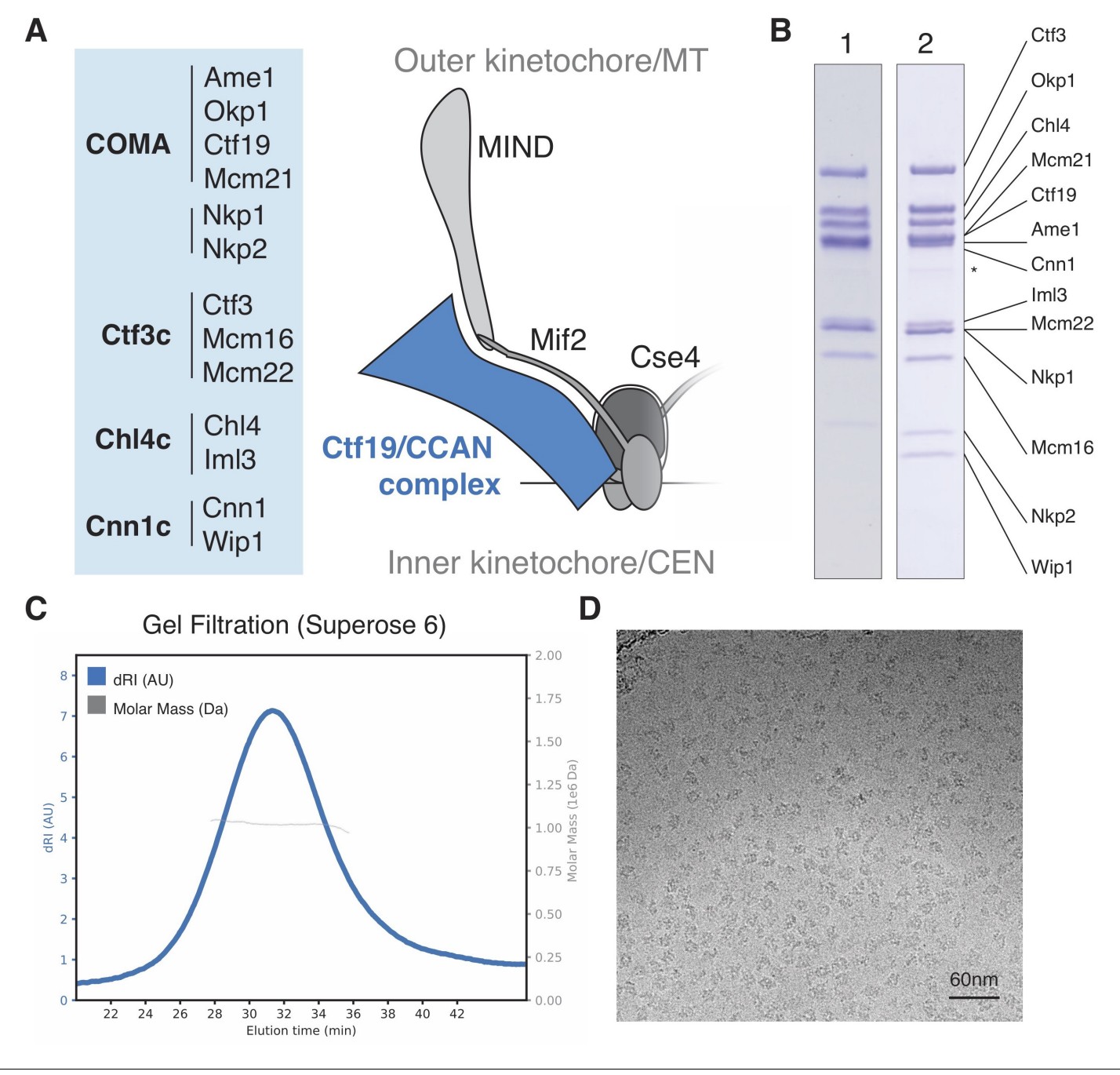

**Figure 1.** Reconstitution of the *S.cerevisiae* Ctf19c. (**A**) List of Ctf19c proteins grouped according to subcomplex and schematic of the yeast inner kinetochore (MT – microtubule; CEN – centromere). (**B**) SDS-PAGE analysis of reconstituted Ctf19c samples (1 – without Cnn1-Wip1; 2 – with Cnn1-Wip1; * – Cnn1 degradation product). (**C**) Mass determination by multi-angle light scattering for the reconstituted Ctf19c used for cryo-EM (dRI – differential refractive index; $M_w$ 1.02 × $10^6$ Da±2.16%). (**D**) Representative micrograph showing Ctf19c particles embedded in vitreous ice. The online version of this article includes the following figure supplement(s) for figure 1:

**Figure supplement 1.** Schematic showing Ctf19c subunits organized by subcomplex membership Members are colored as in *Figure 2C*.

**Figure supplement 2.** Ctf19c preparation and crosslinking procedure.

**Table 1.** Ctf19c/CCAN proteins.

| Complex | H. sapiens | S. cerevisiae | S. pombe |
|---|---|---|---|
| Nucleosome | CENP-A | Cse4 | cnp1 |
| | CENP-C | Mif2 | cnp3 |
| CENP-N/Chl4 | CENP-N | Chl4 | mis15 |
| | CENP-L | Iml3 | fta1 |
| CENP-I/Ctf3 | CENP-I | Ctf3 | mis6 |
| | CENP-H | Mcm16 | fta3 |
| | CENP-K | Mcm22 | sim4 |
| | CENP-M | | |
| COMA | CENP-O | Mcm21 | mal2 |
| | CENP-P | Ctf19 | fta2 |
| | CENP-Q | Okp1 | fta7 |
| | CENP-U | Ame1 | mis17 |
| | CENP-R | | |
| CENP-T/Cnn1 | CENP-T | Cnn1 | cnp20 |
| | CENP-W | Wip1 | wip1 |
| Nkp1/2 | | Nkp1 | fta4 |
| | | Nkp2 | cnl2 |

CENP-C and the Cse4/CENP-A nucleosome. Two copies of each Ctf19c subunit probably decorate each nucleosome core particle (*Weir et al., 2016*). Human CCAN proteins have connectivity and assembly properties similar to those of the yeast Ctf19c proteins (*Table 1*, *Figure 1—figure supplement 1*). Unlike their *S. cerevisiae* counterparts, however, nearly all human CCAN proteins are required for mitosis, while all but two (Okp1 and Ame1) are dispensable in yeast (see Table S1 in *Hinshaw and Harrison, 2018*). Although this difference might imply divergent organizations in yeast and humans, conservation of sequence and domain arrangement for the common components suggests considerable architectural similarity. A lack of structural information for either complex has prevented rigorous assessment of these alternatives.

We present here the structure of the Ctf19c determined by cryo-EM. The structure provides an overview of the inner kinetochore and the coordination of its various functions by showing how Ctf19c components are organized relative to each other, how this organization positions defined Cse4/CENP-A recognition elements, and how the Ctf19c supports regulated recruitment of outer kinetochore proteins that tether the chromosome to the microtubule tip.

## Results

### Reconstitution and structure of the Ctf19c

We assembled the Ctf19c from individual parts (Ame1-Okp1, Ctf19-Mcm21, Nkp1-Nkp2, Chl4-Iml3, Ctf3-Mcm16-Mcm22, and Cnn1-Wip1; *Figure 1A–B*) and purified the recombinant complex to homogeneity by size exclusion chromatography (*Figure 1—figure supplement 2A–B*). Molecular weight determination indicated the presence of two copies of each subunit (*Figure 1C*). Cryo-EM images of the Ctf19c prepared by crosslinking gradient sedimentation showed monodisperse particles (*Figure 1D*), and two-dimensional class averages matched those calculated for an uncrosslinked sample (*Figure 2A*, *Figure 1—figure supplement 2C–D*, *Figure 2—figure supplement 1*). Several class averages resembled projections of a low-resolution tomographic reconstruction of the yeast inner kinetochore (*McIntosh et al., 2013*).

We used these images to determine the structure of the Ctf19c to an overall resolution of ~4.2 Å (*Figure 2—figure supplement 2*). A twofold symmetry axis in the calculated density map related equivalent sides, which were separated by a central cavity (*Figure 2B*). Secondary structure

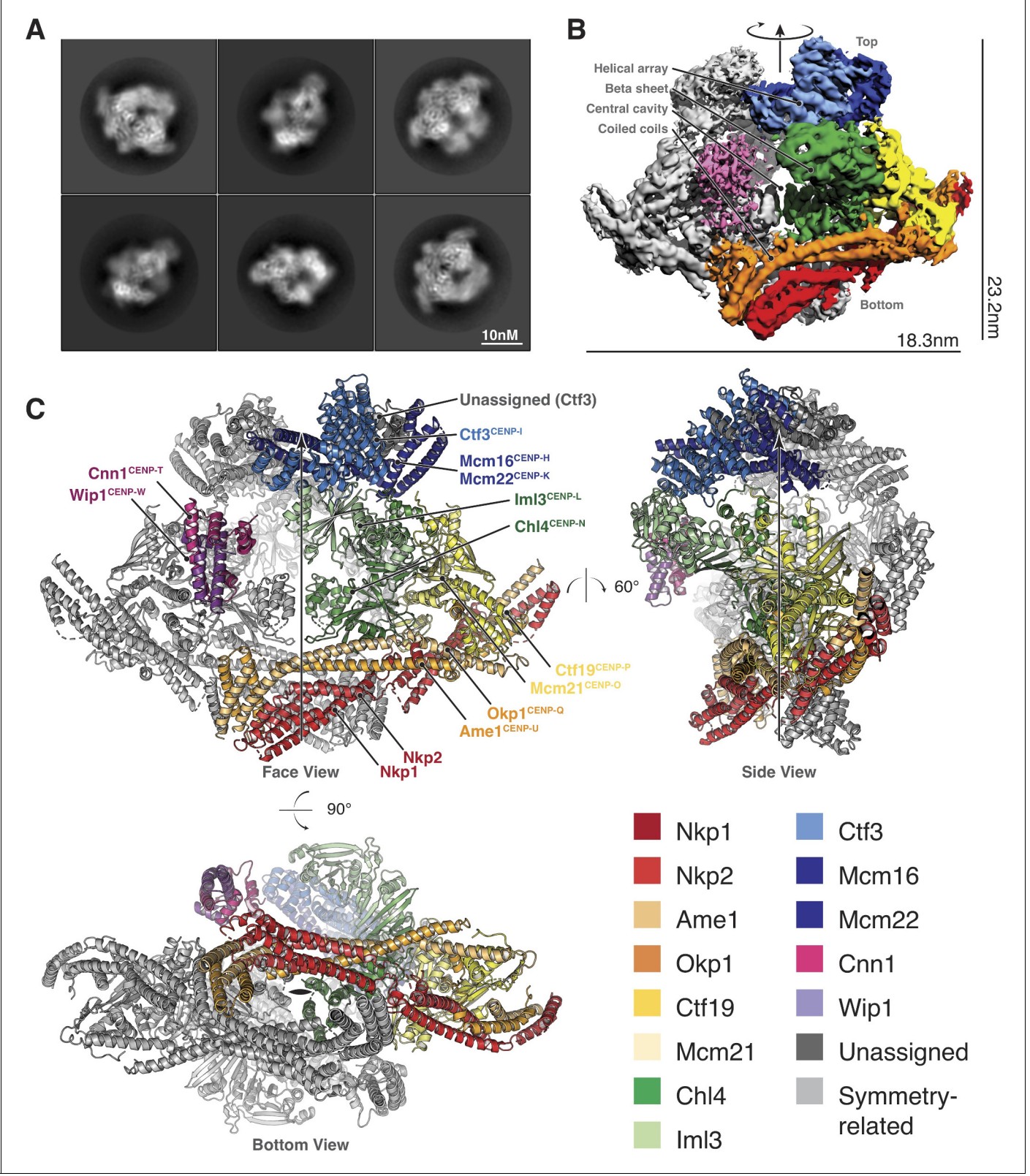

**Figure 2.** Structure determination and model of the Ctf19c. (A) Two-dimensional class averages showing various Ctf19c projections. (B) Initial Ctf19c density map with twofold symmetry applied. (C) Molecular model of the Ctf19c. The twofold symmetry axis is marked by an arrow. Subunits from one monomeric assembly are colored according to their identities. Those related by twofold symmetry are colored gray. The face view corresponds to the orientation shown in panel B.

*Figure 2 continued on next page*

*Figure 2 continued*

The online version of this article includes the following figure supplement(s) for figure 2:

**Figure supplement 1.** Initial three-dimensional cryo-EM reconstructions of the Ctf19c.
**Figure supplement 2.** Data processing summary for Ctf19c structure determination.
**Figure supplement 3.** Model summary and examples of map quality.

predictions and published crystal structures enabled the assignment of Ctf19c subunits to the density (*Figure 2C*). An extended helical network, which matches secondary structure predictions for Ame1, Okp1, and Nkp1/2, spans the bottom of the map. Atomic coordinates describing the Mcm21-Ctf19 dimer from *Kluyveromyces lactis* (*Schmitzberger and Harrison, 2012*) fit into density at the side of the map, with the N-terminal extensions of both proteins projecting towards the helical array at the top of the map. A cluster of five short alpha helices constitute a domain that reaches into the central cavity, and this density matched a crystal structure of the N-terminal domain of CENP-N (*Pentakota et al., 2017*), the human ortholog of Chl4. A crystal structure of Chl4$^{374-450}$-Iml3 from budding yeast (*Hinshaw and Harrison, 2013*) fit into the remaining beta sheet density. The helical array in the top part of the map corresponds to the Ctf3 complex (Ctf3c), which contains the Ctf3, Mcm16, and Mcm22 proteins. Mcm16 and Mcm22 associate as a coiled-coil that traverses the predicted HEAT repeats in Ctf3 (*Basilico et al., 2014*). Cnn1-Wip1, which form a heterodimeric histone fold complex, were visible at the tip of the Ctf3 helical array. From bottom to top, this organization matches published recruitment dependencies for Ctf19c proteins (*Pekgöz Altunkaya et al., 2016*); subcomplexes at the bottom of the map are required for recruitment of their partners towards the top.

A refined map corresponding to a single Ctf19c protomer (see Materials and methods) showed amino acid side chain density for much of the complex, guiding modeling of individual polypeptide chains and their interactions (*Figure 2—figure supplement 3*). For the Ctf3c and for parts of COMA that are not well-resolved, we fit the density with poly-alanine chains and numbered the residues according to their approximate positions. The final model is consistent with published data, our own biochemical observations, and secondary structure predictions, all of which contributed to our assignment of the density to the constituent parts.

The structure shows that, rather than a network of binary interactions, the Ctf19c/CCAN is a defined complex in which subunits interdigitate. Several subunits project N-terminal extensions that are disordered in our reconstruction and that support regulated interactions with other kinetochore components. We describe this model as it relates to published biochemical and structural information in the following sections.

## Ctf19-Okp1-Mcm21-Ame1 (COMA) and Nkp1/2

The conserved four-protein COMA complex connects inner and outer kinetochore proteins. Published work shows that the Ame1 and Okp1 subunits interact with DNA, Mif2, and Chl4, although precise contacts have been difficult to define (*Hornung et al., 2014*; *Schmitzberger et al., 2017*). An N-terminal extension of the Ame1 protein recruits outer kinetochore proteins by making an essential contact with the MIND complex (*Hornung et al., 2014*). Ctf19 and Mcm21 have regulatory functions and recruit Ctf19c components downstream of Ame1-Okp1 (*Pekgöz Altunkaya et al., 2016*). The Nkp1/2 heterodimer stabilizes COMA (*Schmitzberger et al., 2017*), but neither protein is needed for viability or to recruit other Ctf19c members to the complex (*Cheeseman et al., 2002*; *Pekgöz Altunkaya et al., 2016*).

The organization of Ame1-Okp1 and Nkp1/2, which interact in pairs near their N termini and all four together near their C termini, resembles that of the MIND complex (*Figure 3A*, *Figure 3—figure supplement 1*, *Dimitrova et al., 2016*). Two N-terminal four-helix bundles, comprising helical hairpins of Ame1 and Okp1 (analogous to head II of MIND) and Nkp1 and Nkp2 (analogous to head I), are distinct but adjacent 'head' domains. Intermediate segments of Okp1, Ame1, and Nkp1 contact the C-terminal RWD domains of the Ctf19-Mcm21 heterodimer. C-terminal parts of Okp1 Ame1, Nkp1 and Nkp2 form a parallel, four-chain, helical coil. The crystal structure of Ctf19-Mcm21 bound to a fragment of Okp1 helped define the sequence register and orientation of Okp1 early in map interpretation (*Schmitzberger et al., 2017*). Published hydrogen-deuterium exchange

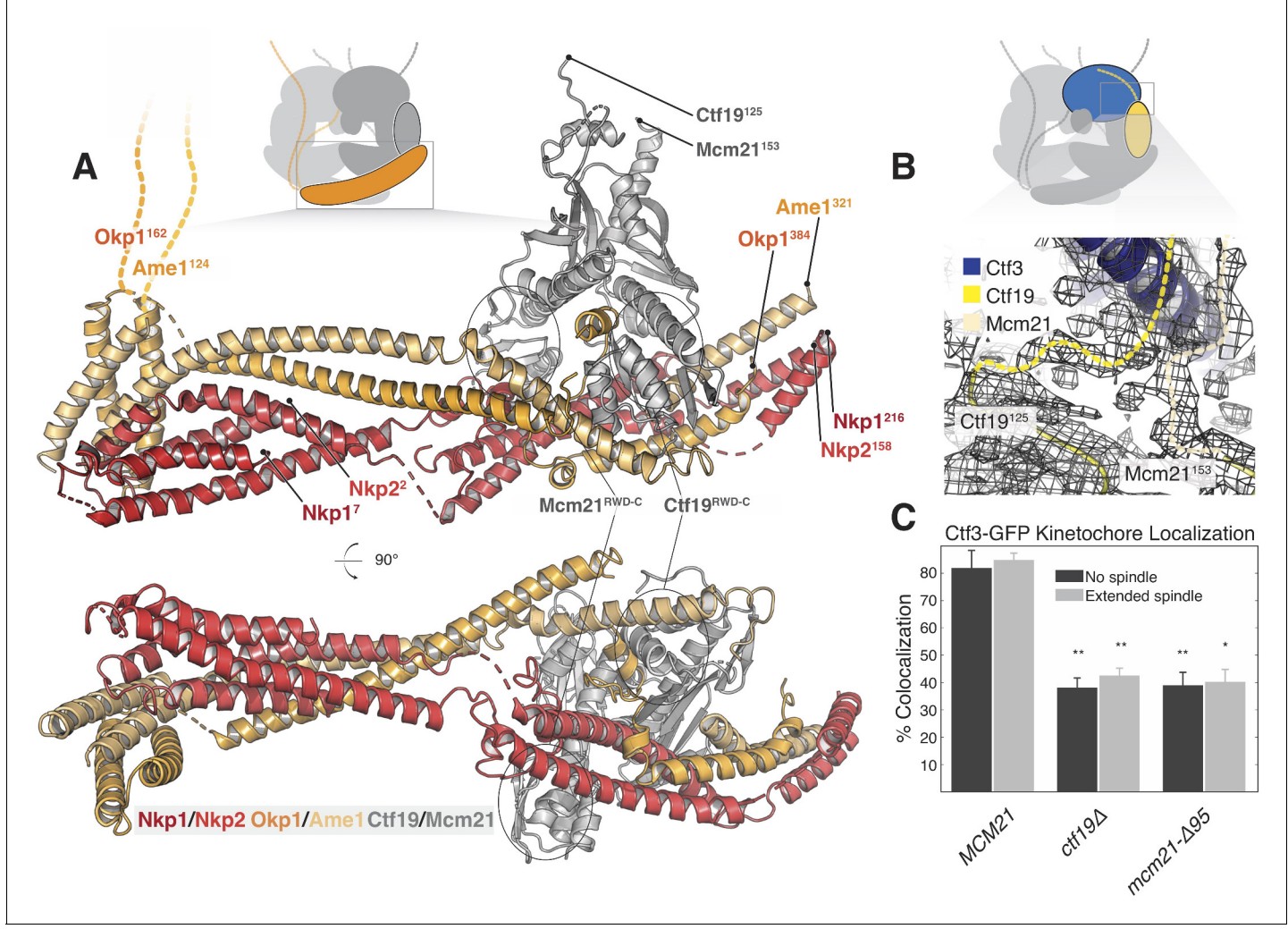

**Figure 3.** Structure of the COMA complex and implications for regulation and Ctf3 recruitment. (**A**) Two views of the COMA complex. The Nkp1/2 complex is colored red, Ame1-Okp1 is orange, and Ctf19-Mcm22 is gray. (**B**) Density for the N-terminal extensions of Mcm21 and Ctf19. (**C**) The Mcm21 N-terminal extension is required for Ctf3 localization. Cells from the indicated strain backgrounds expressing Ctf3-GFP and Mtw1-mCherry were imaged during asynchronous growth, and Mtw1-mCherry foci were scored for colocalized Ctf3-GFP foci (* – p<.005, ** – p<0.001, Student's t-test versus *MCM21*, two tails, unequal variance).

The online version of this article includes the following source data and figure supplement(s) for figure 3:

**Source data 1.** This directory contains tracking data for Ctf3-GFP imaging experiments.
**Figure supplement 1.** Comparison of MIND and COMA structures.
**Figure supplement 2.** Analysis of Mcm21 and Okp1 N-terminal extensions.
**Figure supplement 3.** Fluorescence microscopy images of Ctf3-GFP-expressing cells.

experiments confirmed chain identities for the extended helices (*Schmitzberger et al., 2017*). Positions of chemical crosslinks allowed assignment of much of the corresponding peptide sequence (*Hornung et al., 2014*). Unchanged hydrogen-deuterium exchange within the Mcm21 C-terminal RWD domain in the presence or absence of Ame1-Okp1 (*Schmitzberger et al., 2017*) corroborated the assignment to Nkp1 of density that snakes across the Mcm21 RWD surface. Connections between the Ctf19-Mcm21 C-terminal RWD domains, which are recurring structural modules in the kinetochore (*Schmitzberger and Harrison, 2012*), and Ame1-Okp1 recall previously-described RWD interactions (*Corbett and Harrison, 2012*; *Petrovic et al., 2014*; *Schmitzberger et al., 2017*).

Ctf19 and Mcm21 have N-terminal extensions that are flexible in the absence of other components (*Schmitzberger and Harrison, 2012*). We now see density for parts of these segments that connect the ordered parts of Ctf19-Mcm21 to the Ctf3c (*Figure 3B*). Protein sequence alignments

predict that the N-terminal extension of Mcm21 has a conserved helix at its tip (*Figure 3—figure supplement 2A*). Density near the C-terminal region of the Ctf3 solenoid, which we modeled with alpha helices with no clear chain assignment, likely accommodates this structural feature of Mcm21. To test the idea that this fragment engages the Ctf3c and influences its recruitment to the kinetochore, we imaged cells expressing Ctf3-GFP as they progressed through the cell cycle (*Figure 3C*, *Figure 3—figure supplement 2B*, *Figure 3—figure supplement 3*). Deletion of the Mcm21 N-terminal extension (*mcm21-Δ95*) produced defective Ctf3 localization, an effect that was similar in magnitude to that produced by *CTF19* deletion. In both *ctf19Δ* and *mcm21-Δ95* cells, we observed residual Ctf3 localization, a phenotype also observed in *chl4Δ* cells (*Pot et al., 2003*). We infer that Ctf19-Mcm21 recruits the Ctf3c through Mcm21$^N$ and that additional interactions with either COMA or Cse4-Mif2 support partial Ctf3 localization in the absence of Ctf19-Mcm21.

## Chl4-Iml3

The Chl4-Iml3 complex is heterodimeric with two functional domains (*Hinshaw and Harrison, 2013*). The Chl4 N-terminal domain binds DNA, while the Chl4 C-terminal domain associates with Iml3 to make an extended beta sheet that recruits other Ctf19c components to the kinetochore. Vertebrate CENP-A recognition by CENP-N/L depends on contact between the CENP-N β3-β4 loop and the CENP-A RG loop (*Chittori et al., 2018*; *Pentakota et al., 2017*). An N-terminal bundle of five alpha helices, which forms a pyrin domain (*Pentakota et al., 2017*), contributes to the DNA binding activity of the protein. Our density map showed that yeast Chl4 shares these structural features (*Figure 4A*) and also enabled modeling of the Chl4 linker domain, which connects the two previously-described functional modules and provides extensive contacts between Chl4 and Ctf19-Mcm21 (*Figure 4B*). The map also showed density corresponding to the β3-β4 loop, which extends into the Ctf19c central cavity and contacts the Ame1-Okp1 coiled-coil (*Figure 4C*). Iml3 contacts the Ctf3c (described below), and its exposed beta sheet surface, which is positively charged, faces the central cavity and is therefore well positioned to complement the negatively charged phosphate backbone of nucleosomal DNA.

## Ctf3-Mcm16-Mcm22-Cnn1-Wip1

The Ctf3 trimer (Ctf3, Mcm16, and Mcm22) and the Cnn1-Wip1 dimer form a complex that recruits the microtubule-binding Ndc80 complex to the kinetochore through a flexible N-terminal extension of Cnn1 (*Bock et al., 2012*; *Pekgöz Altunkaya et al., 2016*; *Schleiffer et al., 2012*). In addition to contact with Mcm21-Ctf19, which is described above, Ctf3 contacts Iml3 through a network of bulky residues in both proteins (*Figure 2—figure supplement 3D*) that fixes the position of the Ctf3c relative to Iml3. A recent crystal structure of a chimeric Ctf3 complex confirms the orientation of the Ctf3 peptide and the position of the Mcm16 and Mcm22 C-terminal regions in our model (*Hu et al., 2019*). The crystallized sample lacks the N-terminal regions of Mcm16 and Mcm22, but an extended peptide occupies the cavity formed by the Ctf3 HEAT repeat domain, taking the place of the parallel coiled-coils seen in the *S. cerevisiae* complex.

Density located above the N-terminal extensions of Ctf3, Mcm22, and Mcm16 and adjacent to the Ctf19c central cavity accommodates the histone-fold domains of a Cnn1-Wip1 heterodimer (*Figure 2—figure supplement 1*). Cnn1 contacts Ctf3 through a flexible N-terminal extension and a 'histone fold extension' motif (*Pekgöz Altunkaya et al., 2016*), suggesting an orientation for Cnn1-Wip1 that would position the Cnn1 N-terminal extension over Ctf3. The published observation that CENP-T/W likely interacts with DNA linking CENP-A and adjacent nucleosome particles (*Takeuchi et al., 2014*) agrees with this organization. As its links to other Ctf19c proteins are flexible peptides, Cnn1-Wip1 is unlikely to have a fixed orientation in the Ctf19c in the absence of the Cse4 nucleosome and flanking DNA, thus accounting for the low resolution of the corresponding region in our map.

## Implications for nucleosome-Mif2 recognition by the Ctf19 complex

The vertebrate CCAN interacts with CENP-A through a 'CENP-C signature motif' in CENP-C and the N-terminal domain of CENP-N (*Carroll et al., 2009*; *Kato et al., 2013*). Hydrophobic residues near the C-terminus of CENP-A interact with CENP-C, and the yeast Cse4 C-terminal tail has a similar hydrophobic character (*Kato et al., 2013*), implying a conserved mechanism for CENP-A/Cse4

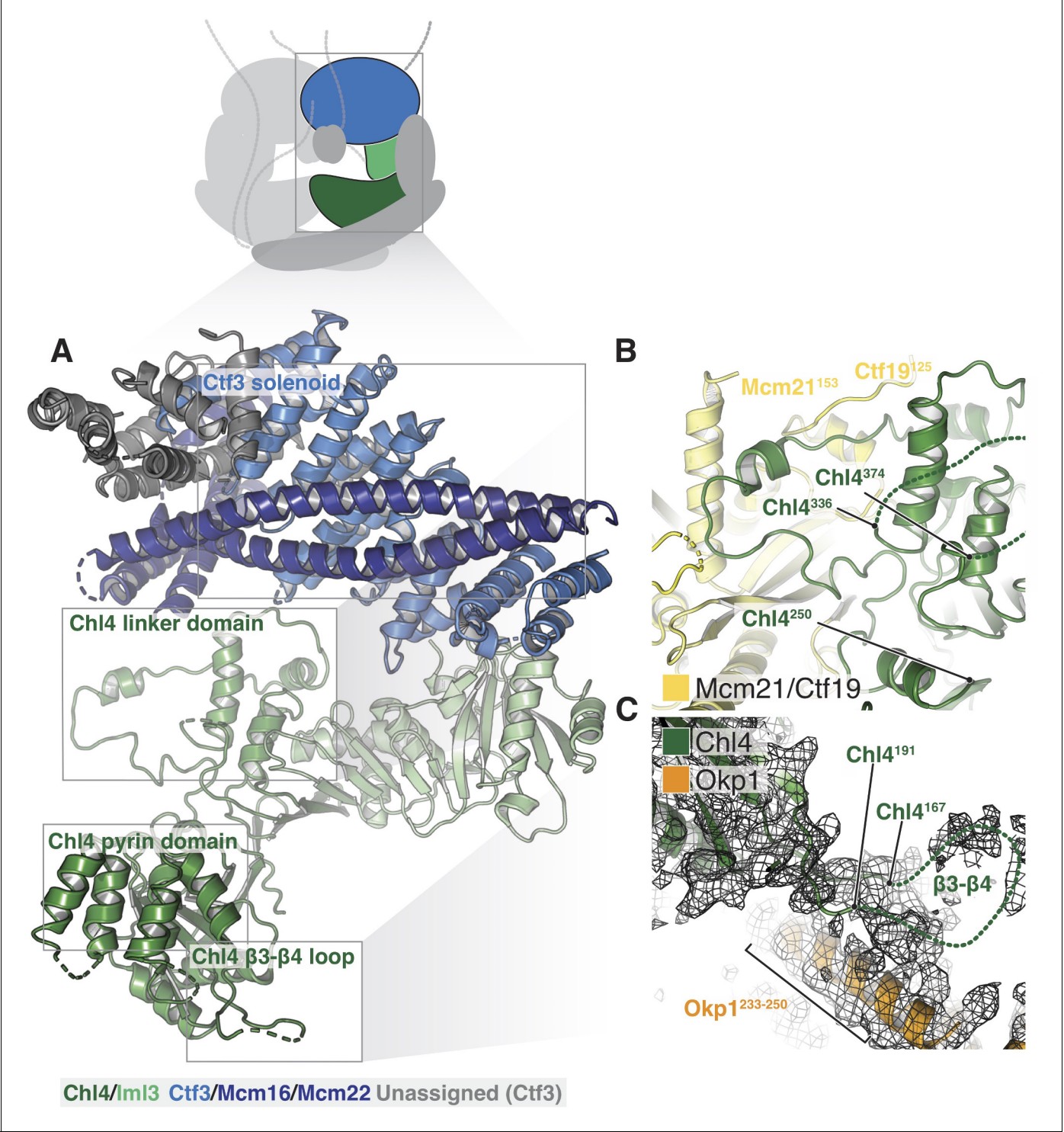

**Figure 4.** Structure of Chl4-Iml3 and the Ctf3c. (**A**) Overview of the Chl4-Iml3 and Ctf3-Mcm16-Mcm22 complexes. Individual domains are indicated. The Chl4 pyrin domain resembles the human CENP-N pyrin domain (PDB 6EQT, *Pentakota et al., 2017*). (**B**) Close-up view of the interaction between the Chl4 linker domain and Ctf19-Mcm21. (**C**) Close-up view of the Chl4 β3-β4 loop with the corresponding map region shown.

recognition. Similarly, the N-terminal domains of vertebrate CENP-N and yeast Chl4 have nearly identical overall folds (*Pentakota et al., 2017*). The current structure accounts for these two CENP-A/Cse4 contact points.

Mif2/CENP-C interacts with the MIND complex, the Ctf3/CENP-I complex (in human cells), Chl4/CENP-N, Cse4/CENP-A, the Ame1-Okp1 dimer, and itself (through a cupin fold homodimerization motif) (*Figure 5A*) (*Carroll et al., 2010*; *Cohen et al., 2008*; *Dimitrova et al., 2016*; *Hinshaw and Harrison, 2013*; *Hornung et al., 2014*; *Klare et al., 2015*). Mif2/CENP-C-interacting regions of the Ctf19c in the current reconstruction allow us to trace the likely path of Mif2 (*Figure 5B*), positioning the N-terminal fragment of the peptide above the top part of our map. In humans and in yeast, this fragment interacts with MIND once Ipl1/Aurora B has phosphorylated Dsn1 (*Dimitrova et al., 2016*; *Petrovic et al., 2016*). Its placement in our model near Cnn1[N], another recruiter of the microtubule

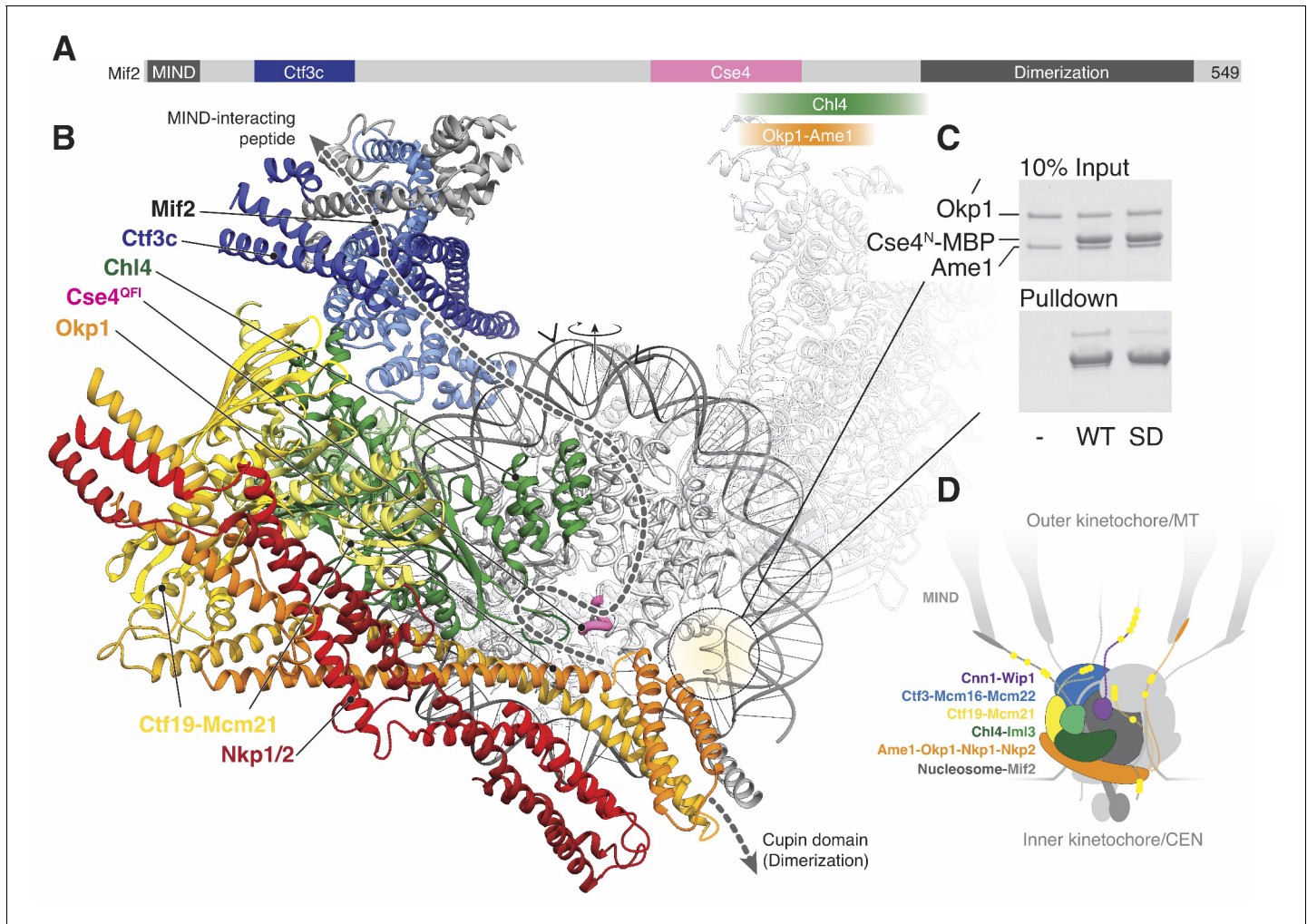

**Figure 5.** Implications for regulated Cse4 nucleosome recognition by the Ctf19c. (**A**) Schematic of the Mif2 protein with segments that interact with other kinetochore proteins labeled and colored according to the model in panel B. (**B**) Model for a monomeric Ctf19c engaging the Cse4/CENP-A nucleosome (PDB 6C0W). The Ctf19c subunits are colored as indicated in *Figure 2C*. A second Ctf19c, in a position related by the twofold symmetry axis of the nucleosome, is shown with transparent ribbons. Mif2 is modeled as a dashed line that satisfies the interactions shown in panel A. Arrowheads above the modeled DNA indicate the approximate boundaries of CDEII according to the phasing of CEN3 DNA on a reconstituted Cse4 nucleosome particle (*Xiao et al., 2017*). (**C**) Ame1-Okp1 binds the Cse4 N-terminal extension. Cse4[1-50]-MBP was incubated with Ame1-Okp1 before MBP pulldown with amylose resin (WT – Cse4[WT]; SD – Cse4[S22D, S33D, S40D]). (**D**) Model of the Ctf19c-Mif2-Cse4 nucleosome complex. Ctf19c proteins are colored as in *Figure 2C*. Approximate positions of phosphorylation sites are marked by yellow circles.

The online version of this article includes the following figure supplement(s) for figure 5:

**Figure supplement 1.** Schematic depicting rearrangements required for Ctf19c-nucleosome assembly.

interaction apparatus (*Huis In 't Veld et al., 2016*; *Malvezzi et al., 2013*; *Nishino et al., 2013*), indicates a collection of extended phosphopeptides likely forms the regulated interface for spindle microtubules (*Figure 5D*). We have not modeled observed contact between CENP-N[C]-CENP-L and CENP-C (Chl4[C]-Iml3 and Mif2 in yeast) (*Pentakota et al., 2017*; *Weir et al., 2016*), because we did not observe an equivalent contact in our reconstitutions of the yeast proteins (*Hinshaw and Harrison, 2013*).

Cryo-EM structures of CENP-N decorating the CENP-A nucleosome (*Chittori et al., 2018*; *Pentakota et al., 2017*; *Tian et al., 2018*) orient the nucleosome with respect to our model. Superposition of the Chl4 N-terminal pyrin and CENP-L/N homology domains in our monomeric Ctf19c model with the same domains in human CENP-N (*Pentakota et al., 2017*) shows that the Cse4 nucleosome would contact the concave surface of a Ctf19c protomer (*Figure 5B*), an arrangement that requires disruption of the Ctf19c dimer contacts observed in the twofold-averaged reconstruction. The Cse4 N-terminal segment would extend towards Ame1-Okp1, and the N-terminal four-helix bundle of Ame1-Okp1 would contact the AT-rich CDEII region of the yeast centromere, accounting for the finding that Ame1-Okp1 binds DNA (*Hornung et al., 2014*). A published genetic interaction between Ame1-Okp1 and the Cse4 N-terminal region (*Boeckmann et al., 2013*) and our observation that Cse4[N] and Ame1-Okp1 must be situated near each other led us to ask whether Ame1-Okp1 interacts directly with Cse4. We found, consistent with recent findings from others (*Anedchenko et al., 2019*; *Boeckmann et al., 2013*; *Halwachs et al., 2018*), that Cse4[1-50] bound recombinant Ame1-Okp1 (*Figure 5C*). Phospho-mimetic mutations in this region of Cse4 (Cse4[1-50]-S22D, S33D, S40D) weakened the interaction. Because cells bearing either phopsho-null or -mimetic Cse4[N] substitutions are viable (*Boeckmann et al., 2013*), Cse4 phosphorylation likely regulates the ability of either Cse4[N] or Ame1-Okp1 to recruit other factors, possibly including Sgo1 (*Mishra et al., 2018*).

## Discussion

The reconstruction we present here shows the overall organization of the yeast Ctf19c, enables assignment of amino acid side chain positions for much of the complex, and suggests a model for its engagement with the Cse4/CENP-A nucleosome.

### Regulated and hierarchical recruitment of kinetochore components

Assembly of the Ctf19c in vivo is hierarchical, with removal of one factor disrupting recruitment of downstream components (*Lang et al., 2018*; *Pekgöz Altunkaya et al., 2016*). When oriented as in *Figure 2C* (face view), the locations of Ctf19c proteins from bottom to top correspond to the published recruitment hierarchy. For example, *ctf19Δ* cells do not efficiently localize Iml3 or Ctf3 to kinetochores, while deletion of either *CTF3* or *IML3* does not affect Ctf19 localization (*Pot et al., 2003*). Nkp1 and Nkp2 are exceptions; they are not required for Ctf19c assembly (*Pekgöz Altunkaya et al., 2016*) but are positioned at the base of the complex. Examination of the Ctf19c model shows that Nkp1/2 removal would indeed not weaken interactions between remaining Ctf19c proteins. Recruitment dependencies for human CCAN proteins differ from their Ctf19c counterparts (*Musacchio and Desai, 2017*), but a low-resolution single particle reconstruction shows that the overall shape of the complex is conserved (*Pesenti et al., 2018*). This may reflect differing relationships between individual CCAN and Ctf19c components, a different orientation of the full complex relative to centromeric chromatin, or interactions between adjacent CCAN modules, each built upon a distinct CENP-A nucleosome foundation. Further structural studies will distinguish among these possibilities.

### Orientation of phosphopeptides for regulation of kinetochore functions

Kinase activities, which rise and fall during the cell cycle, converge on the inner kinetochore to regulate its assembly via phosphorylation of N-terminal extensions of Ctf19c proteins. Four of the Ctf19 complex proteins included in our reconstitution are known phosphoproteins in vivo. These are Cnn1, Ctf19, Ame1, and Okp1 (*Bock et al., 2012*; *Hinshaw et al., 2017*; *Holt et al., 2009*; *Schleiffer et al., 2012*). Mps1, Cdk1, and Aurora B phosphorylate the N-terminal part of Cnn1 to regulate its interaction with the MIND (at least in vertebrates) and Ndc80 complexes (*Bock et al., 2012*; *Huis In 't Veld et al., 2016*; *Malvezzi et al., 2013*). DDK phosphorylates Ctf19 as cells enter

S phase, generating a binding site for the cohesin loading complex (*Hinshaw et al., 2017*). Apposition of Ctf19 and Ctf3 in our model shows how the Ctf19c coordinates DDK recruitment and activity. N-terminal extensions of Mif2, Ame1, Okp1, Cnn1, and Ctf19, although not themselves visible in our map, would be positioned as a cluster of flexible peptides that are available to factors that approach the kinetochore from the direction of the corresponding spindle pole. With the exceptions of Ctf19 and Cnn1 (*Bock et al., 2012*; *Hinshaw et al., 2017*; *Malvezzi et al., 2013*), the contributions of these phosphopeptides to chromosome segregation have not yet been characterized.

## Nucleosome core particle accommodation

In order to accommodate a Cse4/CENP-A nucleosome, the dimeric Ctf19c particle we report must either undergo a dramatic conformational rearrangement or dissociate into monomeric halves (*Figure 5—figure supplement 1A*). Both the presence of monomeric particles in our uncrosslinked Ctf19c preparations and the published observation that, in solution, recombinant COMA-Nkp1/2 visits both monomeric and dimeric states (*Schmitzberger et al., 2017*) support the second possibility, with dimer dissociation occurring by disruption of Okp1-Nkp1/2 contacts. The N-terminal parts of Okp1, Ame1, and Cse4 are kinase substrates (*Boeckmann et al., 2013*; *Holt et al., 2009*), and all are well-positioned to regulate conversion between monomeric and dimeric forms of the Ctf19c.

The yeast centromere is a 125 bp DNA sequence with three conserved motifs: CDEI, CDEII, and CDEIII (*Fitzgerald-Hayes et al., 1982*), with the 80 bp CDEII in the central turn of the histone-associated DNA (*Xiao et al., 2017*). Our model for Cse4 nucleosome recognition generates a clash between Iml3 and CDEI and III, which flank the central turn (*Figure 5—figure supplement 1B*). The clash could be resolved in one of two ways. First, the orientation of the yeast nucleosome relative to the Ctf19c might be sufficiently different from the orientation determined by superposing the Chl4 N-terminal domain onto the corresponding CENP-N domain in recent cryo-EM structures (*Chittori et al., 2018*; *Pentakota et al., 2017*; *Tian et al., 2018*) so that the alternative orientation would accommodate fully wound nucleosomal DNA to either side of CDEII. The Chl4 β3-β4 loop, which is 26 residues longer than its human counterpart, might account for such a difference. Second, partial unwinding of the centromere from the histone octamer, leaving just CDEII in contact with the histone proteins, could also resolve the clash (*Figure 5—figure supplement 1C*). In this scenario, CDEI and III would be positioned near each other as the DNA exits the nucleosome particle, enabling contact between Cbf1, which binds CDEI, and CBF3-Ndc10, which binds CDEIII (*Cho and Harrison, 2011*). The reconstituted Cse4 nucleosome favors partial DNA unwrapping (*Dechassa et al., 2011*), implying that the geometry of the Cse4 nucleosome might present a favorable substrate for Ctf19c assembly. In either case, extensive and distributed contacts between the centromeric nucleosome and the Ctf19c we have presented here can explain the finding that CENP-A is particularly resistant to removal from chromatin (*Cao et al., 2018*; *Guo et al., 2017*; *Smoak et al., 2016*), a characteristic that solidifies centromere identity.

## Materials and methods

**Key resources table**

| Reagent type (species) or resource | Designation | Source or reference | Identifiers | Additional information |
|---|---|---|---|---|
| Gene (*S. cerevisiae*) | See *Supplementary file 3* | | | |
| Strain, strain background (*S. cerevisiae*) | S288c | | | |
| Genetic reagent (*S. cerevisiae*) | See *Supplementary file 3* | | | |
| Antibody | anti-FLAG-HRP (mouse monoclonal) | Sigma | A8592 | (1:1000) |
| Antibody | anti-PGK1 (mouse monoclonal) | Invitrogen | 459250 | (1:5000) |
| Antibody | goat anti-mouse-IgG-HRP (rabbit polyclonal) | Abcam | Ab97046 | (1:10000) |

*Continued on next page*

*Continued*

| Reagent type (species) or resource | Designation | Source or reference | Identifiers | Additional information |
|---|---|---|---|---|
| Recombinant DNA reagent | See *Supplementary file 4* | | | |
| Cell line (*T. ni*) | High Five cells; *Trichoplusia ni* | ThermoFisher | B85502 | Harrison lab stock |
| Cell line (*E. coli*) | Rosetta 2(DE3) pLysS; *E. coli* | EMD Millipore | 71403 | Harrison lab stock |
| Chemical compound, drug | Glutaraldehyde | Sigma | G4004 | |
| Software, algorithm | UCSF Image4 | *Li et al., 2015* | | |
| Software, algorithm | SerialEM | *Mastronarde, 2005* | | |
| Software, algorithm | MotionCor2 (v1.1.0) | *Zheng et al., 2017* | | |
| Software, algorithm | CTFFIND4 (v4.1.8) | *Rohou and Grigorieff, 2015* | | |
| Software, algorithm | Relion (v2.1) | *Kimanius et al., 2016* | | |
| Software, algorithm | Eman2 (v2.22); e2initialmodel.py | *Tang et al., 2007* | | |
| Software, algorithm | ResMap (v1.1.4) | *Kucukelbir et al., 2014* | | |
| Software, algorithm | PyMol (v2.1.0) | Schrödinger, LLC | | |
| oftware, algorithm | Chimera (v1.11.2) | *Pettersen et al., 2004* | | |
| Software, algorithm | Coot (v0.8.8) | *Emsley et al., 2010* | | |
| Software, algorithm | Phenix (v1.13) | *Afonine et al., 2018* | | |
| Software, algorithm | TrackMate (v3.0.0) | *Tinevez et al., 2017* | | |
| Software, algorithm | MAFFT | *Katoh et al., 2017* | | |
| Software, algorithm | JalView | *Waterhouse et al., 2009* | | |
| Software, algorithm | Phyre2 | *Kelley et al., 2015* | | |
| Software, algorithm | Fiji | *Schindelin et al., 2012* | | |
| Software, algorithm | python 2.7.2 | www.python.org | | |
| Other | C-flat | Electron Microscopy Sciences | CF-1.2/1.3–3C | holey carbon grids |

## Protein expression and purification

Ctf19c members were purified either from *Escherichia coli* (Ame1-Okp1, Ctf19-Mcm21, Ctf3-Mcm16-Mcm22, Chl4-Iml3, Nkp1/2) or *Trichplusia ni* (Cnn1-Wip1) cells overexpressing the His-tagged recombinant proteins. The Ame1-6His; Okp1 expression plasmid used in this study codes for Ame1 from *S. cerevisiae* strain YJM1355, which differs from S288c as follows: L97P and G269E. Neither residue is explicitly modeled in the deposited structure. The plasmid also lacks the codon for the final amino acid residue of Okp1 (H406). For expression in *E. coli*, cells were grown to an optical density of ~0.5, and protein expression was induced by addition of IPTG (0.4 mM final concentration). Cultures were then incubated overnight at 18 °C before harvesting by centrifugation and freezing in buffer D800 (20 mM HEPES, pH 7.5, 800 mM NaCl, 10 mM imidazole, 2 mM ß-mercaptoethanol, 10% glycerol by volume) for bacterial cells (~6 mL/L of culture) or B100 (D800, but with only 100 mM NaCl) for insect cells (~10 mL/L of culture) at −80 °C. Protease inhibitors aprotinin, leupeptin, pepstatin, and PMSF were added immediately before freezing.

Protein complexes were purified as described previously (*Hinshaw and Harrison, 2013*). Cell pellets were thawed, supplemented with protease inhibitors as above, treated with ~1 mg/ml lysozyme (*E. coli* expression only), and sonicated for two minutes. After lysis, soluble material was recovered by centrifugation for 30 min at 18,000 rpm in a Beckman JA-20 rotor. Proteins were purified from this extract by means of $Co^{2+}$ affinity chromatography. After elution from the $Co^{2+}$ resin, proteins were applied to a 5 ml ion exchange column (GE HiTrap Q HP: Ctf19-Mcm21, Cnn1-Wip1, Nkp1/2; GE HiTrap SP HP: Ame1-Okp1, Ctf3-Mcm16-Mcm22, Chl4-Iml3) equilibrated in buffer B100 and eluted by an eight-column volume gradient into buffer D800. Purification tags were removed by incubation with TEV protease for two hours at room temperature before removal of cleaved 6His

tags and protease by $Ni^{2+}$ chromatography. We did not remove tags from Ame1-Okp1 or Cnn1-Wip1. For MBP fusion proteins (Cse4-MBP), Co2 +column eluate was concentrated by ultrafiltration without the ion exchange and tag removal steps. All protein samples were further purified on a Superdex 200 column (10/300 GL, GE) equilibrated in gel filtration buffer (20 mM Tris-HCl, pH 8.5, 200 mM NaCl, 1 mM TCEP). Peak fractions were collected, concentrated by ultrafiltration, frozen in gel filtration buffer with 5% glycerol by volume, and stored at −80 °C until use.

## Ctf19 complex assembly and purification

Frozen protein complexes were thawed and mixed at an equimolar ratio for one hour on ice before further purification. For size exclusion chromatography, assembled complexes (250 picomoles in ~40 μL) were applied to a Superose six column (PC 3.2/30, GE Healthcare) equilibrated in gel filtration buffer supplemented with 0.02% sodium azide. For purification by gradient centrifugation, we followed an approach that had previously been reported for eukaryotic RNA polymerase purification (*Schilbach et al., 2017*). Complexes were assembled in gel filtration buffer (500 picomoles in ~80 μL total volume) and were then layered on top of a 5 mL continuous glycerol gradient (10–35% by volume; 80 mM KoAc, 20 mM HEPES, pH 8.5, 1 mM TCEP), the bottom of which contained. 1% glutaraldehyde by volume. After centrifugation for 18 hr at 33,000 rpm in a Beckman SW50.1 rotor at 4 °C, gradient fractions were recovered by bottom puncture, and glutaraldehyde was immediately quenched by mixing with 10 mM aspartate, 20 mM lysine (1:10 by volume). After a 10 min incubation on ice, fractions were subjected to two rounds of dialysis against Tris-acetate buffer (80 mM KOAc, 40 mM Tris-HCl, pH 8.5, 1 mM TCEP) at 4 °C (12 hr and 2 hr) to remove glycerol. Fractions were then concentrated by ultrafiltration at room temperature before application to glow-discharged grids for screening by cryo-EM.

## Multi-angle light scattering

Ctf19c samples were prepared as described (crosslinking glycerol gradient) and analyzed by size-exclusion chromatography coupled to multi-angle light scattering after dialysis to remove excess glycerol and glutaraldehyde. For size exclusion chromatography, we used a 3 ml Superose 6 gel filtration column (GE) equilibrated in gel filtration buffer supplemented with 0.02% sodium azide. The column eluate was passed directly to a Wyatt tReX refractometer for absolute refractive index determination and subsequently to a Wyatt Helios II light scattering detector. Data were processed according to standard pipelines implemented in the Astra software package (Wyatt).

## Cryo-EM sample preparation and imaging

Ctf19c samples were applied to glow-discharged C-flat grids (CF-1.2/1.3–3C; Electron Microscopy Sciences). In all cases, 3.5 μL of protein solution were applied, and grids were blotted from both sides for 4 s before vitrification in liquid ethane using a Cryoplunge 3 instrument (Gatan) operating at 80–90% humidity. For screening of sample preparations and generation of initial maps, we used a Tecnai F20 (FEI) microscope operating at 200 kV. Images, collected using the UCSF Image4 software package (*Li et al., 2015*), were recorded on a K2 Summit electron detector (Gatan) operating in super-resolution movie mode (50 frames, 0.2 s/frame,~60 electrons per $Å^2$ total dose, 0.64 Å/super-resolution pixel).

For collection of high-resolution data, we used an FEI Polara microscope (FEI) operating at 300 kV. Images, collected using the SerialEM software package (*Mastronarde, 2005*), were recorded on a K2 Summit electron detector operating in super-resolution movie mode (40 frames, 0.2 s/frame, 52 electrons per $Å^2$ total dose, 0.615 Å/super-resolution pixel). In total, we collected 15,439 movies over three sessions.

## Cryo-EM image processing

For data collected on both F20 and Polara microscopes, movie frame processing was carried out in MotionCor2 (*Zheng et al., 2017*). Patch-corrected (5-by-5) and dose-weighted averaged movies were used for subsequent steps except determination of contrast transfer function parameters for each micrograph, which was performed on unweighted summed images using CTFFIND4 (version 4.1.8; *Rohou and Grigorieff, 2015*). The pixel size was set to the physical pixel size of the detector by binning in reciprocal space in MotionCorr2. Initial two-dimensional class-averages were

constructed from a manually-picked set of ~50,000 particles (F20, uncrosslinked sample lacking Cnn1-Wip1). A subset of these average images was selected and used as a reference for particle picking in all reported experiments. Particle picking and subsequent steps, except where described otherwise, were carried out using Relion 2.1 (*Kimanius et al., 2016*). Filtering information beyond 20 Å, ignoring CTF correction until the first peak in the picking procedure, and optimizing the particle picking parameters yielded particle sets that, upon examination of the original images, were not biased towards specific views or particle compositions. An initial three-dimensional model was also generated using these two-dimensional averages and the program e2initialmodel.py (*Figure 2—figure supplement 1B*; *Tang et al., 2007*). For the initial model, C2 symmetry was enforced.

To generate a high-resolution reconstruction, we first extracted particles from the dose-weighted summed micrographs collected at 300 kV (Polara data) and binned these stacks in reciprocal space to a pixel size of 2.92 Å. We selected good particles by two rounds of two-dimensional classification and subjected particles from the good classes to three-dimensional classification using an ~11 Å density map calculated from data collected at 200 kV (F20 images of crosslinked particles, *Figure 2—figure supplement 1C*) filtered to 60 Å resolution as a starting model. After classification into six classes, particles from the single best class were chosen and centered by re-extraction from the dose-weighted micrographs at a pixel size of 1.23 Å (the physical pixel size of the detector). Particle sets from separate data collection sessions were pooled at this point and subjected to two-dimensional classification. Most particles partitioned into well-resolved classes, and these were subjected to further three-dimensional classification. Refinement of the best class of particles, using the classification result as a reference and invoking two-fold symmetry, yielded a map resolved to ~4.7 Å (gold-standard FSC criterion, *Scheres and Chen, 2012*).

In order to account for variations in the angle relating the two Ctf19c protomers and variations in the orientation of the top and bottom parts of the map relative to each other, we performed signal subtractions and masked refinements as described in *Figure 2—figure supplement 2B*. Particles from the refined best class described above were subjected to the following operations: symmetry expansion about the z-axis (to map all protomers to a single half-map volume), real-space signal subtraction to isolate signal corresponding to the mapped half-volume, and masked refinement of these half-particle images. The resulting map was resolved to an overall resolution of ~4.4 Å, although Ctf3 density was poorly defined. These operations were performed using subtraction masks that either included or excluded Cnn1-Wip1 density.

To improve the Ctf3-containing part of the density map, we again performed real space signal subtraction to remove density corresponding to the bottom of the map as it is displayed here. This modified particle stack, which represents density corresponding only to the Ctf3c and Iml3, was subjected to three-dimensional classification without refinement. A single best class emerged. Particles belonging to this class were selected, and a corresponding particle stack containing a full Ctf19c protomer was used for three-dimensional refinement. After B-factor sharpening, the resulting map showed high-resolution features throughout much of the density, and the HEAT repeats of Ctf3, along with the Mcm21-Mcm16 coiled-coil was well-defined. This map was used for late stages of model building and for model refinement. Local resolution for this map was calculated using ResMap (*Kucukelbir et al., 2014*). Finally, three-dimensional refinement of the subtracted particle stack corresponding to just the Ctf3c and Iml3 enabled visualization of helical density that was poorly resolved in larger maps due to flexibility relative to the core of the complex, an observation supported by two-dimensional class average images (*Figure 2A*).

## Model building and refinement

We used maps resulting from B-factor sharpening at different levels in order to see side chain density (where visible) and overall connectivity (*Supplementary file 1*). We also aligned and compared model fits to maps corresponding to different steps in our processing procedure. We docked crystal structures of individual components into the density using Chimera. These included Ctf19-Mcm21-Okp1$^{319-342}$ from *K. lactis* (PDB 5MU3), Chl4$^{374-450}$-Iml3 from *S. cerevisiae* (PDB 4JE3), and human CENP-N$^{1-213}$ (6EQT). Except for Chl4$^{374-450}$-Iml3, each of these models required modification of the peptide backbone and reassignment of the primary sequence to match the *S. cerevisiae* versions. This was carried out in Coot (*Emsley et al., 2010*) with the aid of multiple sequence alignments compiled using MAFFT (*Katoh et al., 2017*) and visualized using JalView (*Waterhouse et al., 2009*). We

used multiple sequence alignments to determine the endpoints of conserved secondary structure elements and confirmed these assignments using large side chain densities in the map.

Density corresponding to several large aromatic side chains suggested possible sequence registers for Ctf3 helical repeat domain. However, the absence of a higher resolution crystal structure made modeling of precise amino acid positions unreliable, and we therefore modeled the Ctf3c as an alanine trace with the exception of helical segments abutting Iml3. Residue numbering corresponds to the expected positions of secondary structure elements.

We modeled into the remaining helical density using phenix.find_helices_and_strands (*Terwilliger, 2010*). The resulting model required extensive rebuilding and chain reassignment, which we carried out in *Coot* (*Emsley et al., 2010*). We used secondary structure predictions generated by Phyre2 (*Kelley et al., 2015*) and multiple sequence alignments compiled in MAFFT to guide model-building. We also compiled model restraints from published crosslinking-mass spectrometry, hydrogen-deuterium exchange, and biochemical reconstitution experiments (*Chittori et al., 2018*; *Hornung et al., 2014*; *Klare et al., 2015*; *Pekgöz Altunkaya et al., 2016*; *Schmitzberger et al., 2017*; *Weir et al., 2016*). We used phenix.secondary_structure_restraints to generate initial secondary structure restraints, which we then modified and used to refine the model using phenix.real_space_refine (*Afonine et al., 2018*). All model building and refinement was carried out with a model corresponding to a Ctf19c protomer, and the dimer structure was constructed by fitting component protomers into the C2-averaged map.

*Supplementary file 2* contains a list of Ctf19c subcomplexes, models used as templates, procedures undertaken for model construction, and modeled residues. Segments modeled as poly-alanine, which are also listed in *Supplementary file 2*, correspond to regions where amino acid sequence assignment to the peptide density was not possible. In other regions, especially the N-terminal regions of Ame1-Okp1 and Nkp1/2, we have assigned sequence according to the most likely register, paying attention to amino acid conservation, large amino acid side chains, and the hydrophilic character of helix surfaces. Amino acid side chains were clearly visible for most of Chl4, Ctf19, and Mcm21. Iml3 was well-described by a previous crystal structure (PDB 4JE3). Model building for the Ctf3c is described above. For *Figure 2C*, the structure of chicken CENP-T/W (PDB 3B0C, *Nishino et al., 2012*) was docked into the Cnn1-Wip1 density in our map, and the atomic coordinates for this model were not refined against our map. Figures were prepared with Chimera (*Pettersen et al., 2004*) and PyMol (v2.1.0, Schrödinger, LLC).

## Live-cell microscopy and image analysis

Yeast cultures were propagated in synthetic complete medium (SC, Sunrise Science) prior to imaging. Cells were immobilized on cover slips that were pre-coated with concanavalin A (Sigma) before imaging on an inverted Nikon Ti2 fluorescence microscope with Perfect Focus System and a Nikon Plan Apo 60 × 1.4 NA oil-immersion objective lens. The stage temperature and humidity were controlled with a Tokai Hit stage top incubator set to 30˚C. At least four stage positions were chosen for each strain, and all strains shown for a given experiment were imaged on the same slide during the same imaging session. Images were collected on a Hamamatsu Flash4.0 V2 +sCMOS camera using NIS-Elements Image Acquisition Software. For each stage position, images were taken at 9 z-heights, each separated by. 35 µm, and image stacks were collected at 8 min timepoints for at least 90 min (Ctf3-GFP) or 60 min (Ctf19-GFP). Illumination and frame times were kept constant between experiments.

To analyze Ctf3-GFP images, we first calculated maximum intensity projections in the z direction. Movies were then segmented separately in the mCherry and GFP channels using TrackMate for Fiji (*Schindelin et al., 2012*; *Tinevez et al., 2017*). Segmentation settings were established for the wild-type strain and were subsequently applied to all samples without adjustment. All segmentation results were visually inspected to avoid segmentation artifacts. Measurements for all spots were written to files which were subsequently parsed and plotted. Mtw1-mCherry spots separated from their nearest neighbor by greater than 10 µm were counted as 'No spindle' observations. Histograms showing distributions of measured spindle lengths are shown in *Figure 3—figure supplement 2B*. For this panel, 'No spindle' observations were assigned a 0 µm inter-kinetochore distance. Image statistics from all stage positions from a given strain and experiment were pooled, while statistics from distinct experiments (imaging sessions) were kept separate and compared. Error bars shown indicate standard deviations for measurements from three distinct experiments.

## Yeast growth conditions and western blot

Yeast cultures were grown in a shaking incubator set to 30 °C. Strains were constructed by integration of PCR products using standard methods (*Longtine et al., 1998*). GFP-tagged strains are derivatives of those from the GFP set (*Huh et al., 2003*). Antibodies used for Western blot were as follows: anti-FLAG-HRP – Sigma A8592; anti-PGK1 – Invitrogen 459250; goat anti-mouse-HRP – Abcam Ab97046.

## Data and materials availability

The cryo-EM reconstruction is deposited in the Electron Microscopy Data Bank (EMD-0523). The Ctf19c coordinates are deposited in the Protein Data Bank (PDB 6NUW).

## Note added in proof

Newly-determined cryo-EM density maps suggested an alternative numbering scheme for the Ctf3 protein that would accommodate an N-terminal domain not visualized in the published density and coordinates. This modification does not change the main conclusions of the paper. The Ctf3 residue assignments in *Figure 2—figure supplement 3*, panel 7, have been removed - in consultation with eLife editors - during production to reflect this new information.

## Acknowledgements

We thank Zongli Li and Melissa Chambers for cryo-EM support, Ruben Diaz-Avalaos and Niko Grigorieff for helpful discussions and advice, and Simon Jenni for advice during model building and for providing PyMol scripts. We thank Florian Schmitzberger, Yoana Dimitrova, and Trisha Davis for providing plasmids used in this work and Catherine Maddox for providing GFP-tagged yeast strains. Simon Jenni, Florian Schmitzberger, and Adèle Marston provided valuable feedback on the manuscript before publication.

## Additional information

### Funding

| Funder | Author |
| --- | --- |
| Howard Hughes Medical Institute | Stephen M Hinshaw<br>Stephen C Harrison |
| Helen Hay Whitney Foundation | Stephen M Hinshaw |

The funders had no role in study design, data collection and interpretation, or the decision to submit the work for publication.

### Author contributions

Stephen M Hinshaw, Conceptualization, Resources, Data curation, Formal analysis, Funding acquisition, Validation, Investigation, Visualization, Methodology, Writing—original draft, Project administration, Writing—review and editing; Stephen C Harrison, Conceptualization, Resources, Funding acquisition, Project administration, Writing—review and editing

### Author ORCIDs

Stephen M Hinshaw (iD) http://orcid.org/0000-0003-4215-5206
Stephen C Harrison (iD) https://orcid.org/0000-0001-7215-9393

### Decision letter and Author response

Decision letter https://doi.org/10.7554/eLife.44239.sa1
Author response https://doi.org/10.7554/eLife.44239.sa2

## Additional files

### Supplementary files

• Supplementary file 1. Data and model statistics. Tables describing cryo-EM data collection and molecular modeling.

• Supplementary file 2. Summary of the refined model. Table describing the Ctf19c model. Each subunit is listed along with the number and identities of amino acid residues modeled.

• Supplementary file 3. Yeast strains used in this study (all strains built in the S288C background). Genotypes and names of yeast strains used in this study.

• Supplementary file 4. Protein expression constructs used in this study. Protein expression constructs for Ctf19c subunits are listed.

• Transparent reporting form

### Data availability

We have deposited the model coordinates and cryo-EM maps in the PDB (6NUW) and EMDB (EMD-0523). Tracking files for imaging experiments are included as a source data file associated with Figure 3.

The following datasets were generated:

| Author(s) | Year | Dataset title | Dataset URL | Database and Identifier |
|---|---|---|---|---|
| Hinshaw SM, Harrison SC | 2019 | Model coordinates from The structure of the Ctf19c/CCAN from budding yeast | http://www.rcsb.org/structure/6NUW | RCSB Protein Data Bank, 6NUW |
| Hinshaw SM | 2019 | The structure of the Ctf19c/CCAN from budding yeast | http://www.ebi.ac.uk/pdbe/emdb/EMD-0523 | Electron Microscopy Data Bank, EMD-0523 |

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
