## [Decision Letter]

Thank you for submitting your article "The structure of the inner kinetochore from budding yeast" for consideration by *eLife*. Your article has been reviewed by three peer reviewers, including Andrea Musacchio as the Reviewing Editor and Reviewer #1, and the evaluation has been overseen by Cynthia Wolberger as the Senior Editor. The following individuals involved in review of your submission have agreed to reveal their identity: Stefan Westermann (Reviewer #2); Hongtao Yu (Reviewer #3).

The reviewers have discussed the reviews with one another and the Reviewing Editor has drafted this decision to help you prepare a revised submission.

Summary:

This very important contribution from Hinshaw and Harrison sheds light on the structural organisation of a conserved inner kinetochore complex, known as Ctf19 or CCAN in different organisms. The authors report a cryo-EM reconstruction, at an overall resolution of 4.3 Å, of the *S. cerevisiae*'s Ctf19 complex reconstituted from its sub-complexes. The quality of the map allowed the authors to describe the overall organisation of the complex, to trace several segments of the 13 polypeptide chains that constitute it, and to identify crucial interactions that stabilise the assembly. At present, structural information is available exclusively for small parts of the Ctf19 or CCAN subcomplexes. This landmark study provides a detailed comprehensive view of the complex, filling a substantial gap of knowledge in the kinetochore field and orienting future studies aiming to address the interaction of the inner kinetochore with the specialised Cse4/CENP-A nucleosome. All three reviewers support publication of the manuscript in *eLife* with enthusiasm, but the following major points need to be addressed:

Essential revisions:

1) The authors may elect to test biochemically that the reconstituted yeast inner kinetochore complex is capable of binding to the Cse4/CENP-A nucleosome and the MIND/MIS12 complex. At least the latter should be within their reach. If the former is not available to the authors, it will be desirable to indicate that binding of the reconstituted Ctf19 complex to the Cse4 nucleosome has not been tested experimentally.

2) The dimeric configuration of yeast Ctf19/CCAN complex is intriguing. The authors argue in the Discussion that this dimer needs to be disrupted to accommodate the binding of the nucleosome. Yet, in the model depicted in Figure 5D, a Ctf19 dimer is shown to bind to a nucleosome. Does the Ctf19 dimer need to be completely separated into two halves (as the authors imply) or does it simply need to be partially disrupted while still maintaining some dimer contacts? Can the authors disrupt the dimer interface and show enhanced nucleosome binding activity? Ideally, this issue needs to be clarified through experimentation, but if this is not feasible, they should at least clarify this point through rewriting the paper.

Also, please note that while not being listed as an essential revision, the point below refers specifically to the title of the manuscript, and we would be grateful if you could consider it carefully.

The title could be more specific: "Structure of a 13-subunit CCAN assembly from budding yeast" or something along these lines (as essential parts of the inner kinetochore such as Mif2 or the nucleosome are not revealed in the current study).

---

## [Author Response]

Essential revisions:1) The authors may elect to test biochemically that the reconstituted yeast inner kinetochore complex is capable of binding to the Cse4/CENP-A nucleosome and the MIND/MIS12 complex. At least the latter should be within their reach. If the former is not available to the authors, it will be desirable to indicate that binding of the reconstituted Ctf19 complex to the Cse4 nucleosome has not been tested experimentally.

While we have reconstituted a sample similar to the one described by the Herzog group (Halwachs paper currently posted to bioRxiv and referenced in the main text), we have opted not to describe these experiments in the current manuscript because 1) the reconstituted sample is not of sufficient quality to use for high resolution structure determination by cryo-EM and 2) failure to produce a high-quality reconstitution reflects a yet-incomplete understanding of the underlying biology. Reporting a reconstitution we believe to be incomplete would therefore be slightly misleading.

We have also observed that our Ctf19c reconstitution interacts with MIND/MIS12. Contacts between defined Ctf19c motifs and MIND/MIS12 are described in the literature (Hornung et al., 2016 is one example), and our recapitulation of these contacts does not yet provide new information beyond these findings.

2) The dimeric configuration of yeast Ctf19/CCAN complex is intriguing. The authors argue in the Discussion that this dimer needs to be disrupted to accommodate the binding of the nucleosome. Yet, in the model depicted in Figure 5D, a Ctf19 dimer is shown to bind to a nucleosome.

The Ctf19c bound to the nucleosome (shown in Figure 5D) was generated according to the steps below and does not maintain the dimer contacts shown in Figure 2. We have added Figure 5—figure supplement 1 to clarify this point.

Does the Ctf19 dimer need to be completely separated into two halves (as the authors imply) or does it simply need to be partially disrupted while still maintaining some dimer contacts? Can the authors disrupt the dimer interface and show enhanced nucleosome binding activity? Ideally, this issue needs to be clarified through experimentation, but if this is not feasible, they should at least clarify this point through rewriting the paper.

We generated the nucleosome-containing model in Figure 5D by: 1) creation of a Ctf19c monomer model, 2) superposition of the Chl4 N-terminal pyrin and Iml3-like domains with the corresponding parts of CENP-N from PDB 6C0W, 3) duplication of the entire structure and rotation (180 degrees) about the nucleosomal twofold axis, 4) removal of CENP-N and the redundant nucleosome from 6C0W. These steps are now summarized in Figure 5—figure supplement 1.

Placement of a Ctf19c monomer on the nucleosome particle according to the published CENP-N model precludes the Ctf19c dimer contacts we observe in vitro, leading to our speculation that the dimer must be broken for Ctf19c-nucleosome complex assembly. Ctf19c rearrangement or dissolution (as the case may be) is likely regulated during passage through the cell cycle. Bypass of this presumed regulation may provide access to a sample that does not reflect the underlying biology we wish to capture. Even so, were we able to do so, we would have disrupted the observed dimer by introducing mutations at the interface. Lack of high-confidence amino acid residue assignments in the Okp1-Ame1 and Nkp1-Nkp2 “head domain” peptides, which constitute the main dimer contacts, prevented us from doing this. To emphasize and clarify these points, we have rewritten the relevant parts of the Results and Discussion sections of the manuscript.

Also, please note that while not being listed as an essential revision, the point below refers specifically to the title of the manuscript, and we would be grateful if you could consider it carefully.The title could be more specific: "Structure of a 13-subunit CCAN assembly from budding yeast" or something along these lines (as essential parts of the inner kinetochore such as Mif2 or the nucleosome are not revealed in the current study).

We have retitled the manuscript: “The structure of the Ctf19c/CCAN from budding yeast”.